# Co-Preaching: The Effects of Religious Digital Creatives' Engagement in the Preaching Event

Frida Mannerfelt

Faculty of Theology, University College Stockholm, SE-168 39 Bromma, Sweden; frida.mannerfelt@ehs.se

**Abstract:** The preaching event is a complex process of communication and interpretation. The aim of this study is to describe and discuss how the preaching event is affected when it is digitally mediated and involves so-called "religious digital creatives" (RDCs). This is achieved through a case study of the preaching event at two Church of Sweden (CoS) congregations that offered pre-recorded, digitally mediated worship services. The research questions guiding the study were: "When and how do the RDCs engage in the preaching event?" and "How can the effects of this engagement be understood in the light of homiletical theory drawing on the works of Mikhail Bakhtin?" The study found that RDCs engaged in the verbalization phase of the preaching event in several ways—including visualization, direction, editing, enhancement, and contextualization of the sermon—and thus contributed significantly to the preaching event. Furthermore, the RDCs exhibited notable relational authority—an authority based on negotiation, interdependence, and interaction. Employing homiletical theory that draws on Mikhail Bakhtin's work, I argue that the RDCs in this case study are best understood as co-preachers who contribute to expanding the polyphony of the preaching event.

**Keywords:** online preaching; preaching event; homiletics; social media; religious digital creatives; authority; Church of Sweden; Michail Bakhtin; digital mediation

## 1. Introduction

The preaching event is a complex thing.[1] As Wilfrid Engemann (2019, pp. 3–4) has shown, the preaching event is a process of comprehension and communication that consists of several phases of text interpretations and text introductions that involve the interaction between the authors of the Bible text, the Bible, preacher, sermon manuscript, the delivered sermon, listener, and the "auredit" (what the listener has heard), each in their specific context. Therefore, Carina Sundberg (2008, pp. 11–44, 195–99) has argued, the preaching event—as the product of very complex situated interactions between multiple actors[2] like preacher, listener, architecture, liturgy, artifacts and so on —is characterized by "polyagency".

While the preacher, word, and listener are usually the foci of attention in the preaching event, with a few notable exceptions (Kaufman and Mosdøl 2018, pp. 123–32) scant attention has been paid to materiality as an actor in the communication and meaning-making process. However, as practice theorists like Theodore Schatzki has pointed out, all social phenomena are constituted by the entanglement of human practices and material entities, such as bodies and artefacts, and material arrangements like buildings and technology (Schatzki 2019, pp. 19–22).

In her article "Preaching at the thresholds—Bakhtinian polyphony in preaching for children," Linn Sæbø Rystad (2020) argues that materiality is a dimension of preaching that must not be overlooked. She underlines that: "Focusing on materiality might highlight what preaching from a pulpit does or does not do in the communication situation, or which body it is that is preaching" (pp. 122–23). In the article, she discusses the use of "mediational means" (the biblical narrative, costumes, and objects) in preaching for

children. Drawing on James Wertsch, Rystad argues that access to the world is always mediated. For this reason, a scholar should not limit her scope to what humans are doing, but must look into how humans interact with mediating materiality (pp. 45–46, 108–25).

This article will explore and analyze what happens when yet another actor is brought into the complexity of the preaching event: digital technology. According to Schatzki (2019, pp. 19–22, 36–37) human practice has become increasingly dependent on material arrangements enabled by technology, in particular digital technology. Clearly, digital devices are deeply embedded in our daily lives, including worship. In a socio-material perspective, digital technology could be said be an actor in its own right. However, this article will focus on the new human actors that digital technology brings into the preaching event. Dubbed religious digital creatives (RDCs), these are the "individuals whose digital media work and skills grant them unique status and influence within their religious communities" (Campbell 2021, pp. 4–5).

In her book *Digital Creatives and the Rethinking of Religious Authority* (2021), Heidi Campbell (2021) argues that religious authority is transformed by digital media and technology. This transformation is due not only to the transition of established religious authorities (like priests and pastors) from physical spaces into digital environments, but also to the occurrence of new actors (like technicians or social media ministers). They present religious content online and have become religious authorities in their own right. The purpose of this article is to explore and discuss what happens when RDCs engage in a preaching event. This is achieved through a case study of the preaching event in pre-recorded digitally mediated worship services in two Stockholm congregations in the Church of Sweden (CoS).

The research questions guiding this article are: (1) When and how do RDCs engage in the preaching event? (2) How can the effects of RDC engagement in the preaching event be understood? I will argue that the RDCs can be understood as "co-preachers," as they all contribute significantly to the sermon and thus to the preaching event. The effects of co-preaching will be discussed in the light of homiletical theory that focuses on the concept of polyphony.

The article is structured as follows: first, I will present the methodology, material, and the theoretical frameworks employed. In doing so, I will discuss both the concept of RDCs—what it is and how it is applied in the analysis of the article's source material— and the concept of polyphonic preaching invoked in the results discussion. Next, I will describe when and how the RDCs engage in the preaching event. Finally, I will conclude with a discussion of the results in the light of polyphonic preaching, an umbrella term for the Scandinavian line of homileticians inspired by the communication theories of Mikhail Bakhtin.

## 2. Methods, Materials, and Theoretical Frameworks

The case study's source material was gathered within the framework of the action research project *Church in Digital Space*.[3] As part of the project, I collaborated with the New Testament scholar and pastor Rikard Roitto to follow two congregations in the CoS Diocese of Stockholm, Järfälla and Täby, as they developed short, pre-recorded digitally mediated worship services (Mannerfelt and Roitto 2022a, 2022b).

The subject material was created from August 2021 to February 2022, well into the COVID-19 pandemic, and involves six preaching events, three for each congregation. During the period, researchers and practitioners met once a month. The researchers observed the practitioners' preparation for and recording of the worship services, made individual interviews with all practitioners involved, and gathered recordings and screenshots of publication in social media. A month later, practitioners and researchers met for focus group conversations in which the researchers presented an analysis of what they had seen and heard, and theories that could aid the understanding. The researchers also facilitated a discussion in which the practitioners responded to the analysis and reflected on their own practices. Next month, there was a new round of observations and interviews, and so on.

This article, however, uses an ethnographical case study approach to the sources instead of the highly collaborative practices of action research we initially applied.[4] In other words, the practitioners were not involved in the negotiation of research questions or the analysis and presentation of the research except for an opportunity to reflect on the validity of the RDC theory.

The source material thus consists of 6 observation protocols, transcriptions of 18 individual and 6 focus group interviews, 6 edited recordings of the services, and 18 screen shots of how the recordings were presented on the congregation's websites and social media platforms (Youtube, Facebook, and Instagram).[5] In analyzing the source material, I have drawn on Heidi Campbell's work on authority and religious digital creatives.

### 2.1. Religious Digital Creatives

As Campbell and Tsuria (2021, pp. 7–12) point out in the introduction to *Digital Religion: Understanding Religion in a Digital Age*, authority is one of the key research areas and questions in the field of digital religion. In one of her recent books, *Digital Creatives and the Rethinking of Religious Authority,* Campbell pursues the question of what religious authority looks like in an age of digital media. She states that the typical conclusion in scholarly studies of religious authority and new media is that, since digital culture and technology is characterized by features like freedom and a lack of hierarchy, established religious authorities are challenged. In an effort to turn the tables, Campbell (2021, pp. 1–21) asks instead what religious authority looks like and how it is established in a digital context. Her hypothesis is that internet technology and digital culture both facilitate and empower new religious actors, and their wielding of authority creates hybrid structures that over time may change their religious institutions.[6]

To examine how religious authority is structured in a digital culture, Campbell interviewed 120 individuals, all of whom had been active for at least four years and renowned for their digital work for and in Christian churches (ibid., pp. 14, 53). While these interviews took place between 2011–2016, well before the pandemic and the subsequent radical—and rapid—digital transition of churches, they provided a foundational framework for analyzing future digital mediation. Campbell's analysis of the interviews yielded three categories of actors:

1. Digital entrepreneurs, who create digital resources—platforms or content—for their communities in their free time.
2. Digital spokespersons, who are employed to manage a religious community's digital presence.
3. Digital strategists, who already have an official position (e.g., as pastors and deacons), but who use digital media to do their work more effectively.

Common to all three groups is that they possess skills and experience in digital media work—they are "digital creatives"—which gives them unique influence and status in their religious communities. Hence, they are religious digital creatives, RDCs (ibid., pp. 49–53).

The RDCs in question in this study are employed in congregations in Täby and Järfälla, two communities within the CoS's Stockholm diocese. The team in Järfälla consists of a pastor, a religious educator, two technicians (responsible for recording and editing audio and video), and a communications director (responsible for publishing content on digital platforms). The team in Täby consists of a pastor, a deacon, a musician, and a communications director (who records, edits, and publishes content).[7] In other words, the preaching events in this case study included both digital spokespersons (the communications director and tech team) and digital strategists of the online-minister type.

These particular congregations in this study were chosen for several reasons. For starters, the congregations and RDCs are in Sweden, one of the world's most digitalized societies (Digital Economy and Society Index 2022). In addition, the CoS is also one of the world's wealthiest churches, which has allowed congregations to hire employees such as dedicated A/V technicians. In Campbell and Osteen's (2021) study on how churches

digitized during the COVID-19 pandemic, the digital transition was often carried out by either a single pastor (i.e., what Campbell would call "digital strategist") or a small group of volunteers ("digital entrepreneurs"). In this study, we get a glimpse of churches' digital transformation through collaboration between different groups of RDCs. Finally, the RDCs in Campbell's study were mainly focused on missional or educational activities in their digital work. The RDCs in this study are engaged in online worship services and digitally mediated preaching events.

The digital team members at both congregations in this study will be analyzed as RDCs, i.e., people wielding religious authority through their use of digital technology. The analysis will focus on describing what they do, how they understand their work in relation to the digitally mediated preaching event, and what kind of authority they perform through their words and actions.

According to Campbell (2021, pp. 18–37), RDCs use of authority may be described through four categories:

1. Authority as role based (as in the works of Weber)
2. Authority as power struggle (as in the works of Foucault)
3. Authority as relational—where authority is seen as negotiated and mutually beneficial, as described by for example by Mia Lövheim in her study on the authority of bloggers.
4. Algorithmic authority—where algorithms "tells us what voices to listen to, which topics are important and which structures to give weight to in evaluating credibility". (Campbell 2021, p. 31) Algorithmic authority comes from statistics and figures like number of followers, hits and rankings from search engines, or—in an academic setting—the number of publications.

Digital spokespersons tend to describe themselves as institutional identity curators whose task is to present and represent the identity of the community in media, particularly on digital platforms. Sometimes they relate to algorithmic authority, but more often on role-based authority, in particular what Weber called rational-legal authority. In other words, they see themselves as part of a structure with particular rules that they support. Within churches, they do their job to serve church's greater mission. However, in this service they are often caught in something of a contradiction: the same institutions that hired them to do digital media work are reluctant about the use of digital technology. When these digital strategists are called upon by the church's leadership to contribute with their expertise, the shift in power dynamic is not always welcome. Therefore, they tend to be very cautious, and emphasize that their work is not about theological interpretation but about making the message of the church accessible. (Campbell 2021, pp. 110–29, 157–62).

Digital strategists view themselves as missional media negotiators. They work in institutions that claim that they do not need digital technology, but the strategists believe the institutions can do their work more efficiently and creatively with the aid of digital media. They continuously blend online and offline ministry, and see digital platforms as an extension of their local work. In this position, they are bridge-builders who often negotiate. This means that they tend to view authority as relational, as something that is created and negotiated between different parties through communicative interaction. Or, as Campbell summarizes it, "Authority comes to the leader through creating a balanced or interdependent relationship" (Campbell 2021, pp. 133–53, 162–66).

It is worth noting that the digital strategists in this case study differ slightly from Campbell's category because they have not chosen the hybrid role themselves. That is, they were given the task to provide digitally mediated worship services during the COVID-19 pandemic, and they state that they would not have taken on this task if they could avoid it. Over time, however, they have grown into the role as bridge builders between onsite and online church. The fact that their work was sanctioned by the leadership and necessitated by the pandemic might explain why stories of "technological apologetics," the justification narratives that were such a prominent feature of Campbell's pre-pandemic RDCs, are virtually absent from the narratives of these CoS strategists.

These conditions also affect the digital spokespersons in this study. Absent are the narratives so common to digital spokespersons in Campbell's study, namely that the same leadership that hired them to do digital work is also suspicious of digital technology. Also missing are Campbell's accounts of grudges stemming from the shift in authority when the spokespeople are called upon to work as media mentors to for example pastors.

Instead, the strategists in this study express a profound gratitude and trust towards the spokespersons. For example, when asked about one of the recording sessions in which the spokesperson (communications director) clearly was in charge of what, when, and how every part of the worship service should be recorded, the strategist (pastor) said that "in that case, it is [the communications director] who does his thing, he is completely in charge. I gladly let him decide what is best". When asked about this, the communications director himself compared it to the local worship service:

> **(Interviewee):** "[Laughs] If I were to participate in a physical worship service, I would turn to the pastor and musician and ask: "What should I do? Is this right? In what order should this be done?" In the same way, I think they give me more responsibility because they are not at home in this area, even if they have been involved in planning the order of the worship. [ . . . ]

> **(Interviewer):** All right. So in this church space, your "sixth church" [an expression frequently used by the team in Täby about online church as an addition to their five local churches], you are more in charge?

> **(Interviewee):** Yes, you could say that. [Smiles] It is quite exiting that I should know more about a church space in that way.

These quotes are not just examples of how the shifting power dynamics between the usual leaders and the digital spokespersons do not seem to cause unease. They are also examples of the spokespersons' use of relational authority. In Campbell's study, her spokespersons often downplayed their autonomy and personal contribution to the messages and underlined their loyalty towards the theological message of their institutions. In other words, they favored a role-based authority. While the spokespersons in this study in some instances did relate to role-based authority, they also commonly performed and spoke about relational authority.

### 2.2. Polyphonic Preaching

The discussion this study's results builds on the Scandinavian homiletical discussion that draws on the theories of the Russian philosopher and literary critic Mikhail Bakhtin to describe and understand the communication going on in the preaching event. Since the concept of polyphony is central in several of these discussions, I will use the shorthand term "polyphonic preaching" to refer to them.

A landmark volume in the discussion on polyphonic preaching is Marlene Ringgaard Lorensen's *Dialogical Preaching: Bakhtin, Otherness and Homiletics* (Lorensen 2014) in which she explores how preachers expose their preaching to interactions with various 'others' of preaching, and how a Bakhtinian understanding of communication might be incorporated into homiletical theories.[8] While Lorensen herself mainly focuses on Bakhtin's theories of dialogue and carnivalization, the concept of polyphony is intrinsically related to them, and she introduces the concept to offer a theological model of communication for the homiletical strand of "Other-wise preaching" (McClure 2001).

Bakhtin developed the concept of polyphony in his work on Dostoevsky and Rabelais. According to Bakhtin, their novels are dialogical since the characters possess and interact with their own consciousness and voices. As such, the reader does not just hear the author's voice, she also hears the characters' voices and is thus drawn into her own dialogue with them, creating a polyphony. This dialogical polyphony is contrasted with a monological authorship in which the author is omniscient and has the final word on interpretation. To Lorensen (2014, pp. 66–67), "the role of the preacher in contemporary preaching practices has striking similarities to the polyphonic author-position".

In Bakhtin's thinking, communication is thus relational and collaborative. Conversation partners (local and distant) always play a constitutive part in how the speaker develops and shape his or her utterance. With the aid of Bakhtin's theories, Lorensen pleads for a collaborative preaching practice, in which preachers act as hosts who invite others into the conversation and, in the process, become guests themselves. (Lorensen 2011, pp. 26–45; 2014, pp. 66–67). She underlines the importance of the preacher not ventriloquizing different voices, arguing that "If preaching, in spite of its monological appearance, is to function as a dialogical encounter, one of the most important tasks for the preacher, from a Bakhtinian perspective, is to avoid conflating the voices of the listener, preacher, and scripture into one and instead let the polyphony of voices interact in a way that let them transform and enrich each other mutually". To Lorensen, this means that Bakhtin may provide the homiletical movement with "the beginnings of a polyphonic theology of communication" (Lorensen 2011, p. 44).

In the article "Listeners as Authors in Preaching," Gaarden and Lorensen (2013, pp. 28–45) use Bakhtinian perspectives to discuss the empirical findings in Gaarden's study of the listener's meaning-making processes. They argue for a reversed perspective in the analysis of preaching, and challenges fellow homileticians to understand listeners as primary authors of the sermon. They make this rather surprising move in relation to Bakthin's idea that meaning emerges in interaction with dialogue partners. According to Bakhtin, the addressees of an oral or written discourse always play an implicit and explicit part as co-authors, and in this sense the "listener becomes the speaker" (Gaarden and Lorensen 2013, p. 32). Instead of discussing how preachers invite listeners as co-authors in sermon preparation, they want to discuss how listeners invite preachers to be co-authors of their inner reflections during the preaching event. Lorensen elaborates further on this idea in an article written with Gitte Buch-Hansen (Lorensen and Buch-Hansen 2018, pp. 29–41). They argue that the refugees in a Danish church acted as co-authors of practical theology, since they provoked adjustments to the traditional theory of human capital. Furthermore, the refugees' understanding of the ritual challenged traditional Danish Lutheran understandings of the Eucharist and the church. It is this notion of interplay between authors/co-authors that has inspired the concept of "co-preacher" that is found in this study.

In her previously mentioned article, Rystad employs Bahktin's concept of polyphony to analyze two sermons directed to children. Through her discussion about analytical concepts, it becomes clear how communication is tied to authority in Bakhtin's thinking. In delineating both the monological and dialogical, Bakthin makes a distinction between scaffolding (words that are used to build up a monological discourse) and architectonic whole (words that are allowed to influence a dialogical discourse in a way that may lead to transformation and new perspectives). He also makes a distinction between, on the one hand, words that are part of an authoritative discourse that creates monologue, and, on the other, words that are part of and internally persuasive discourse that creates dialogue. In Rystad's interpretation of Bakhtin, authoritative words are words spoken from a distance, which gives an impression of their being more important than our own words, possessing a meaning that must either be accepted or rejected. An inner persuasive word "does not have status or authority and is tightly interwoven with our own words" (Rystad 2020, p. 111). It is creative and interacts with other inner discourses to cause change.

However, Rystad (2020) draws on Olga Dysthe to nuance Bakhtin's notion of authority. Alongside both an authoritarian discourse based on power and tradition, and an inner persuasive discourse free from authority, there is a third discourse of authority based on trust and respect. According to Rystad, preachers often aim for the latter. In her case study, Rystad found that while the sermons started out as polyphonic—particularly through the aid of the mediational means—both sermons ended up as monologues when the preacher stepped in at the end with authoritative words and proclaimed the message of what "all of this truly meant" (p. 45). Rystad concludes that "Polyphony is the most important consideration when laying the groundwork for dialogical interaction with a preaching

event. Polyphony helps create a threshold space in which authoritarian discourses are challenged and narratives re-interpreted" (Rystad 2020, p. 124).

It is no wonder that Rystad makes this move. Authority is not just a key research area in the field of digital religion. Ever since Fred B. Craddock's *As One without Authority* (1971), the question of authority has been at the forefront of homiletics.[9] The issue of authority has been particularly important to homileticians who argue for conversational and/or dialogical approaches, like polyphonic preaching. These scholars tend to trace the development of homiletics and build their argument in relation to authority. As for example the homiletical contribution of John McClure (2001) in his landmark book *Otherwise Preaching* According to McClure,

> preaching is exiting itself through the doors of many deconstructions or gradual otherings. Among these are deconstructions of self, culture, scripture, reason, language, metaphysics, tradition, even of the word itself. Most specifically [ . . . ] preaching is exiting through the deconstructions of the four overlapping authorities that have bequeathed preaching to us: the authority of the Bible, the authority of tradition, the authority of experience, and the authority of reason. (p. 2)

McClure not only launches his homiletical theory in relation to changes in the understanding of authority, but he also writes his overview of the development of the homiletical field to show how homileticians over the years have tried to grapple with the deconstruction of authority. His own solution, which draws on Emmanuel Levinas's idea of "the human other as a site for the revelation of the Holy other," argues for a conversational approach (pp. 47–59).

Authority is also at the center of discussion in Casey Thornburgh Sigmon's thesis "Engaging the Gadfly: A Process Homilecclesiology for a Digital Age" (Sigmon 2017), one of the few longer, in-depth contributions to the homiletical field that specifically engages with preaching in digital culture. According to Sigmon, digital media sheds light on how preaching has been caught in a "pulpit-pew binary," where the pulpit represents the locus of authority and the pew the attentive, silent audience. Sigmon points out that the pulpit and pew easily fall into the dualistic framework in the Western Christian tradition, which justifies one part's dominance over the other. Furthermore, the binary hinges on static, substance-oriented categories often regarded as unchangeable truth (ibid., pp. 5–6).

According to Sigmon, homileticians have tried to solve the problem of the pulpit-pew binary since the 1960's, and she describes a movement towards more relationality and mutuality. However, since the homileticians have not had a clear understanding of the problem, they have not completely solved the problem. In an overview of different approaches to the problem of authority and asymmetric power relations, Sigmon discusses both the New homiletic movement and Other-wise homiletics, as well as feminist, postcolonial, and postmodern perspectives, and while she acknowledges the different tactics to handle the pulpit-pew binary problem, she claims that none of them have actually solved the problem.

She hopes that digital culture will prompt homileticians and preachers to create a preaching event that takes an "exit from the house of the sanctuary," and thus will avoid being delimited by liturgy, architecture, and strictly oral-aural relations. Sigmon underlines that this change does not come about by itself, since digital culture is an algorithmic, capitalist system that can be every bit as problematic as the classic Western binary schema. To avoid the negative effects of digital culture, there need to be "theo-ethical norms" to guide its development. Sigmon draws on process theology to describe such a theology of preaching, calling it a "homilecclesiology" (pp. 16–33).

Especially important to Sigmon in her vision of homilecclesiology is the preaching priesthood of all believers. The contribution of unordained laypeople who lack theological training should define the work of the ordained preachers, whose task it is toto build up the laypeople for the task of interpreting and communicating God's words and actions. The preachers are to model the interpretation of sacred texts and traditioned dogmas in relation to culture through their own words and actions, in particular from the pulpit. Authority ought to be relational, no one should assume power over others, and there can be no imposition of truths and timeless statements (pp. 169–87). Sigmon concludes:

> Rather than seeking to become the authority on everything for the church, we seek to cultivate in the laity a sense of their own authority and capacity to challenge the grasp of unidirectional authorities on their life. [ . . . ] They [preachers] cultivate the ability to affirm, embrace, and expect ever-growing complexity and beauty without losing Christianity's spiritual center and identity among different realities. (p. 185)

She also offers a few suggestions of how this could be done in practice, both on digital platforms (social media) and in hybrid engagement when technology serve to disrupt the monologue from the pulpit (pp. 200–15).

Though Sigmon does not discuss Bakhtinian approaches to preaching in her overview of different homiletical approaches to the "problem" of authority, I would argue that her vision of authority seems to share traits the polyphonic preaching discussion. with Bakhtin's thoughts on a relational authority. The notion of authority described by Lorensen and Rystad seems quite similar, not only to Sigmon's vision of a homilecclesiology suited for a digital culture, but also to the notion of relational authority as described by Campbell's digital strategists and the participants in this case study. In other words, polyphonic preaching is well-suited as a tool for homiletical discussion of the results from exploration of RDCs' engagement in the preaching event.

### 3. When Do the RDCs Engage in the Preaching Event?

In order to envision when the RDCs engage in the preaching event, I will use Engemann's (2019, pp. 1–13) description of the communication and comprehensions processes involved in the preaching event. Engemann divides the preaching event into several phases of text interpretation and text production. First, there is the Phase of Tradition, in which the authors of the Bible interpreted the biblical events and produced the Bible text. Next comes the Phase of Preparation, when the preacher, as author, interprets the Bible texts and produces a sermon manuscript. Then comes the Phase of Verbalization, when the preacher, as sender, interprets his or her sermon manuscript and produces the delivery of the sermon. The final stage, according to Engemann, is the Phase of Realization, in which the listener interprets the delivery of the sermon and produces the auredit (Latin for "what is heard). All these phases takes place in a specific context that contributes to the processed of interpretation and communication. Below is visualization of that process (Figure 1):

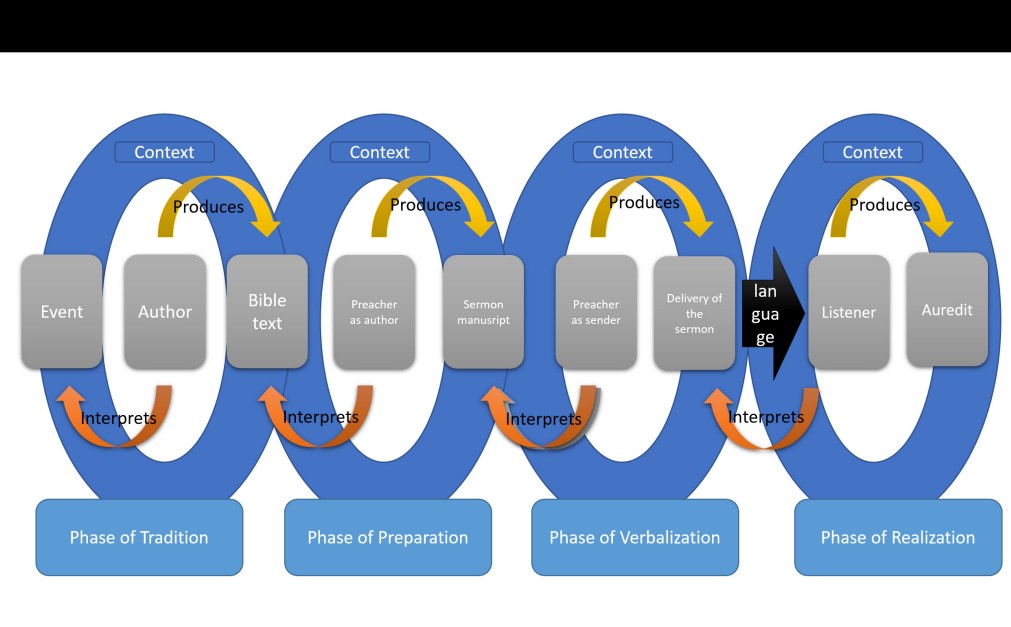

**Figure 1.** Engemann's diagram of the preaching event.

　　　When the RDCs come into the interpretation and production process of the digitally mediated preaching event, the process is affected in several ways. One important factor that decides how the preaching event is affected is which kind of digital technology—the mediational means—is involved. In this study, the sermons were pre-recorded on Tuesdays, edited during the following days, and published at a certain time later in the week (1 PM on Fridays in Järfälla and 10 AM on Sundays in Täby). Both communities published the sermons on the congregation's website and on Facebook, Youtube, and Instagram. When this study's RDCs, with their particular use of digital technology, are inserted into Engemann's diagram (Figure 2), the processes of interpretation and production changes.

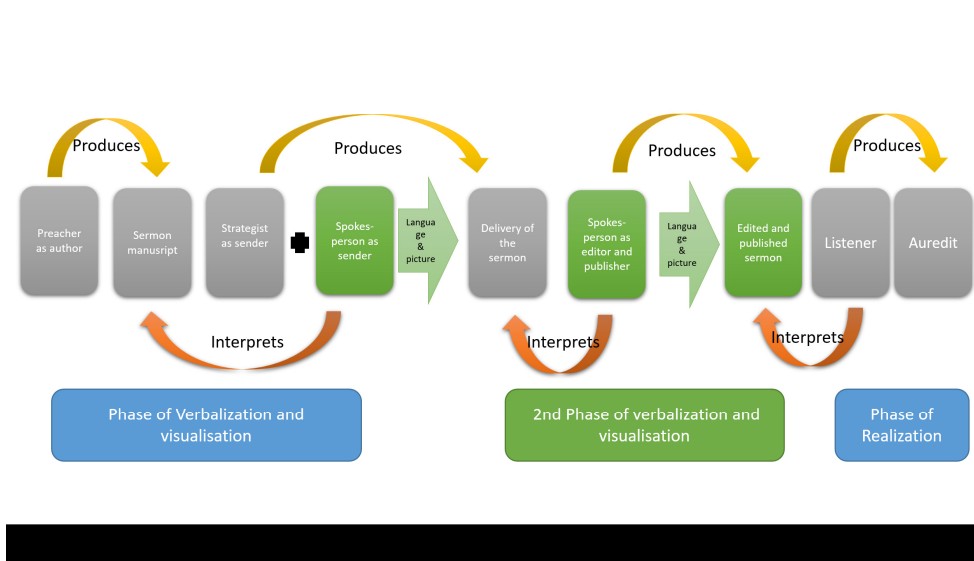

**Figure 2.** The Religious Digital Creatives (RDCs) involvement in the digitally mediated preaching event.

While the phases of tradition, preparation, and realization remain the same in many ways, the phase of verbalization changes. The strategist is no longer the sole interpreter of the sermon manuscript. She is accompanied by the spokespersons who record the sermon. As mentioned above, the technicians are intensely involved in the worship service where the sermon is delivered and, as I will show, directly influence the delivery of the sermon as well as its content.

This change draws attention to how the verbalization phase is not just a phase of the spoken word. It is a phase of verbalization *and* visualization. While it in one sense always has been[10], the visual character of the preaching event is emphasized.

This is in line with an increased emphasis on the visual in contemporary culture. In their overview of visual culture studies, Promey and Brisman (2010, pp. 188–91) show that the notion of contemporary culture as "hypervisual" has grown in importance. They refer to the work by Nicholas Mirzoeff, who argues that contemporary culture has a tendency to picture or visualize experience and create meaning through pictures rather than written words, a tendency linked to the development of digital technology.

Furthermore, in these two cases, a second phase of verbalization and visualization is introduced through the churches' particular use of digital technology. The listeners no longer interpret the delivery of the sermon; instead, they interpret the *edited* version of the sermon. If the listeners access the sermon from a social media platform, they get yet another additional layer of interpretation: the spokespeople's description of the sermon that accompanies and frames the recording. In the following section, how the strategists and spokespeople engage through each phase will be described in detail.

## 4. How Do the RDCs Engage in the Preaching Event?

### 4.1. Engagement in the First Phase of Verbalization and Visualization

The two categories of RDCs considered here engaged in several ways during the first phase of verbalization: through visualization and mediational means, through direction, and through changes to the sermon's content.

As mentioned earlier, during the recording sessions the spokespersons were in charge. They arrived early to the recording location to set up cameras, microphones, lights, and other technical equipment. If the recording took place inside a church, they would adapt the space in different ways to suit their needs. The spokespersons in Järfälla stated that "Everything visual is our responsibility". When asked if they ever discuss the visual with the strategists, they said:

> Some preachers have very clear ideas and thoughts and wishes, and we try to incorporate that if it is possible, suitable, and looks good. But most, in particular pastors, just want to get up and do their thing and do not think about how they stand and how it looks.[11]

They continued to explain how they strive to include the atmosphere from the location, and not just from the preacher's perspective. This was confirmed in the observations. The setting of the recording space varied every time in relation to what the spokesperson thought would catch the atmosphere of, for example, the liturgical year, the theme of the Sunday, or the theme of the sermon (if they had been told what it was in advance). They could also choose a location inside the church that showcased something they thought the participants/viewers would appreciate and meditate on, like a painting or an artifact. Sometimes they chose locations or artifacts that the worshippers would not normally see were they listening from the pew. In other words, the spokespeople were deeply engaged in choosing mediational means intended to interact with the strategists' words.

The spokespersons in Järfälla could also choose a location outside of the church, for example a garden, a square, the cemetery, the children's corner in the parish hall. In those cases, the choice often made in consultation with the strategist. This kind of collaboration with strategists was much appreciated by the spokespersons, since it allowed for creative work and mutual exchange. The spokespersons got inspiration for how the sermon could

be visualized, and they were also able to inspire the strategists. It is worth noting that the authority emerging here, both in words and action, is relational authority.

The spokesperson—the communications director—in Täby, who also doubled as technician during the recording sessions, worked in the same way. He came early to prepare and chose the location, camera angles, and mediational means and to set the scene for what he called "the right atmosphere". During one of the observations, there was a slight dissonance between the spokesperson's choice of location and visualization, and the words of the strategist's sermon. The worship service was to be published on All Saint's Day, and the strategist—who had assumed the recording would take place in the old 13th Century church—had chosen to talk about the life of one of the saints depicted in the medieval paintings in the roof of the church. When the team met on the morning of the recording session, she found out that the spokesperson had chosen a modern church that possessed a very large, beautiful globe for lighting candles. During All Saint's weekend, it is very common for people in Sweden to light candles on graves and in churches, and the spokesperson wanted to feature that in the video. He also wanted to include something about the possibility for people to light a digital candle at the CoS's website. After some negotiation, they agreed that the recording session would take place in the modern church after all, but the picture in question would be added during the editing of the recording. The preacher added a paragraph to the sermon about lighting candles.

Just like his counterparts in Järfälla, the spokesperson in Täby reported that most of the strategists let him take full responsibility for the visualization, but a few strategist's wanted to partake in planning how the sermon and worship service should be envisioned. Likewise, he preferred collaboration since it enabled a creative working environment. He mentioned an example that he was particularly pleased with: a worship service with a pilgrimage theme in which they walked during the recording session. This had required a lot of discussion and negotiation on how he could envision the strategists' words along the path, and how his choice of imagery could be verbalized by the strategist.

The spokespersons also engaged through directions. In both churches, they directed the delivery of the sermon in detail. They told the strategists where to stand, look, how to talk, how to interact with the technology, and even what to wear (for example avoid liturgical clothing in certain situations). When asked about this, the spokesperson in Täby commented that he, in addition to the directions we had observed, often also had to instruct the strategist on style, tone of voice, facial expressions and so on.

When the strategist (pastor) in Täby was asked about her thoughts on these directions (in particular, being told not to wear liturgical clothes), she commented:

> If we are a team and we need to make decisions as a team, and then no one's opinion can be superior. And [the communications director] obviously has a reason for it. Even if I do not understand exactly what it is, I have to let this process grow and see if it turns out well. Perhaps he will say: it turned out the way I wanted, and then I will understand.

Both here and in the example of the negotiation about sermon location, the authority that emerges in words and actions is relational. That authority is negotiated through communicative interaction in an interdependent relationship.

This strategist was not the only one who appreciated directions. On the contrary, according to the spokespeople in both Järfälla and Täby, the strategists often asked for them, especially in the beginning of the transition to digital worship services during the pandemic. The Järfälla spokespeople were sometimes even asked to review the sermon manuscript beforehand. Here, there is a slight difference between the two congregations. As earlier mentioned, the spokesperson in Täby suggested changes to the sermon manuscript that the strategist accepted, an example of relational authority. In Järfälla, the spokespeople stated that they tried to be careful not to control the content of the sermon manuscripts and that they thought it was "very strange to poke around in someone's sermon". They would only weigh in when they were asked to do so.

In other words, the spokespersons' starting point was a role-based authority, one that they were accustomed to from their lengthy service as wardens in the local worship services. However, in the digitally mediated service, they were invited to wield relational authority and consequently did. The Järfälla spokespersons stated that the invitations had been more common in the beginning of the pandemic. They thought the preachers had listened to their feedback and had improved the content of their sermons over time. The spokespersons felt that the strategists had learned to compress their sermons, keeping them short and to the point and delivering them with a personal and casual style.

In sum: the digital spokespersons engaged in the preaching event through visualization of the sermon manuscript, the choice of mediational means, the giving of direction, and the occasional advising on the content of the sermon. In this engagement, there are traces of role-based authority emerging in the spokespeople's narratives, but a relational authority is prominent and emerges in practices and narratives, as enabled and encouraged by the digital mediation.

### 4.2. RDC Engagement in the Second Phase of Verbalization and Visualization: Editing

The engagement of the RDCs in the preaching event created a second phase of verbalization and visualization: editing. In the editing process, it was mainly the spokespersons that were engaged in adding b-roll imagery, texts, and sound.[12] In all cases, the goal of these additions was to enhance or contextualize comment on the message of the delivered sermon.

An example of how the use of b-roll imagery could enhance the delivered sermon can be found in the previously mentioned All Saint's sermon. The spokesperson and the strategist negotiated that the sermon was to be recorded in the modern church, and the painting from the medieval church would be added during editing. When asked about this, the spokesperson thought the editing facilitated the visibility of the painting. Onsite, in the church, the picture was difficult to spot since it was located near the roof at the entrance of the church, and was thus impossible to see if you were seated in the pews. Online, the picture was easier for the viewer to see since the editing included a closeup where the picture's colors and outlines were enhanced. The effect enhancing the painting was that the preacher's words also were enhanced. As the preacher named and explained the saint's particular attributes, the relevant details were highlighted.

The team in Järfälla also frequently added pictures and clips of artifacts, art, surroundings, and other meditational means in the editing process. The spokespersons (technicians) in charge of editing reported that they had an extra hard drive with such material. When asked about how they selected what to include and where to place it, the spokespersons said that they would usually get inspiration after listening to the sermon. They were extremely positive about being able to contribute in this way: "Wow, here we can help and contribute to what they are trying to say through imagery". They gave an example of a pastor who, during Advent, preached on the theme "make way for the Lord," and brought his son's tiny toy car as "prop". The strategist's sermon related to the movie *Cars*, and how the main character, the racing car Lightning McQueen, was sentenced to repair the road he had accidentally destroyed. At first, he tried to do it quickly and sloppily, but then he learned that it was better to do it diligently and slowly. Taking it slow also allowed for detours where Lightning McQueen got to know others including the judge in the town (whom the strategist interpreted as a God figure). The preacher concluded by asking what would be the best way of preparing our hearts for Christmas: fast and expensive, or to take small, slow steps and allow for detours?

To the spokespersons, it had been natural to add a stop-motion animation in the end with the toy car, showing how it drove by slowly, taking a couple of turns. In this way, they wanted to enhance and comment on an element of the sermon that they thought was important.

The spokespersons in Järfälla also stated that they tended to work more with imagery when the preacher had structured their sermon in relation to a metaphor or "prop" of some

kind. In addition, according to them, this practice of bringing in mediational means had increased dramatically among the preachers in the digitally mediated preaching events. In the subsequent focus group conversation, the spokespersons and strategists discussed the reasons for this and concluded that the digital format encouraged the use of mediational means. Interestingly, the reason for this was not just because it was easier to preach from an artifact or art when you were sure that the listeners would actually see it (for example, a tiny toy car would be very difficult to spot from the pews). The strategists also testified that it made them feel that they were less "lonely" in the delivery of the sermon. They saw it as another "body" with which to share the camera's attention.

The spokesperson in Täby worked in similar ways when editing. He added pictures from a variety of shots including imagery that he thought would suit the delivered sermon. Interestingly, in the conversation about this practice, the musician mentioned that she worked in similar ways with her choice of music, for example during funerals. According to her, it happened quite often that she had planned to play certain music during the funeral service, but after hearing the funeral sermon, she changed to something that she thought would enhance or even comment on the message. Both the musician and the pastor in the team commented that they appreciated the communications director's work on imagery, and that they thought it enhanced or even brought new dimensions to the message of the sermon. The pastor, especially, thought it was very interesting "to hear how he thinks in pictures, and how he thinks they [words and pictures] are theologically connected".

Of course, more than pictures could be added during editing. Sound or text could also be added. For instance, in Täby, the music from the hymns could be added to the opening and/or closure of the sermon, which functioned in similar ways as the musician's choice of music during the funeral service: as a contextualization of or enhancement of what the communications director thought was an important message in the sermon. In Järfälla, the communications director captioned the sermons, interpreting the spoken language of the preacher into textual language.

In sum: the digital spokespersons engaged by adding visual, aural, and textual enhancement or contextualization of what the they thought were important parts of the digital strategists' sermon message. The strategists engaged by adapting to the increased visual dimension of the sermon and the spokespersons' directions. Notably, the collaboration between these particular strategists and the spokespersons was characterized by negotiation and trust. While there were no instances when, for example, the choice of b-roll imagery obscured or contradicted the strategists words, it could potentially have occurred since the strategists did not review the spokespersons' choice of additions before publication in digital media. The spokespersons' interpretation and idea about what messages are important to enhance or convey is even clearer in the publication part of the second phase of verbalization and visualization.

### 4.3. RDC Engagement in the Second Phase of the Verbalization and Visualization Process: Publishing

When publishing the delivered, edited sermon, the spokespersons (communications directors) followed the rules of engagement in social media: they chose a thumbnail, a small image representation of the content of the recording of the worship service. They also wrote a short accompanying text—often a summary of the theme or content of the sermon—to encourage those who encountered the sermon on the church's website or social media channels to watch the video. In this way, yet another layer of interpretation and production was added to the process. Notably, the core messages presented in social media were not always the same as the preachers' core message in the sermon itself. Take, for instance, the All Saint's sermon which was previously mentioned.

As discussed, the spokesperson wanted to pay attention to the practice of lightning candles on graves and in churches during the All Saint's feast. The strategist adapted her sermon to accommodate that wish, and the sermon was recorded with the preacher standing next to the candleholder in the back of the church. This was mirrored in the publication phase. The main part of the sermon was the strategist's original sermon where

she spoke about the saint as an example of how having Jesus as a light in your life, and how living your life with a firm hope of paradise, can affect your whole life. However, the sermon's framing in social media did not mention that at all. Instead, it included a thumbnail picture of the preacher next to the globe of light, with a text that read: "During the All Saint's weekend we remember those who have passed away. On [link to website] you can light a digital candle for someone you miss and watch it burn alongside candles lighted by other people. You are not alone in your grief".

A second example can be found in the Järfälla church. In this case, the sermon was on John Chapter 4. The strategist started by asking if the listener had heard about "the woman at the well who met Jesus," and painted a picture of a woman who was cast out and living in shame. However, when she met Jesus she became a "living advertisement poster" for Jesus. It was not an advertisement as in trying to sell a product, but:

> She advertises because she is deeply touched by what he has said. He tells her about her life and it is true. She responds in honesty, bares herself. She dares to stand there with her shame and meets the man who will totally change her life. She receives the living water and wants to pass it on to others. The story is about us. We get to be living advertisement posters, here and now, with a message about love. A message that goes beyond what we can think and imagine, where there is no room for shame and self-loathing, and we are surrounded by love, grace, and mercy, where we can live in love, here and now, and pass it on to the people we meet, just like that woman went to others and told them about the meeting with the man who had told her everything.

In social media, the communications director wrote: "Maybe shame can't get a hold on someone who has been seen for who they truly are? Reflect together with [pastor] who tells the story about the woman at the well in the first digital worship service of the year. From St Luke's church".

The message about passing love on and our calling to preach a message of love in our everyday life was omitted. Instead, the spokesperson chose to emphasize the part about shame and being seen. In this case, there is a shift in content. Instead of the strategist's message about how Jesus changed the woman's life and wiped away her shame (sin) with grace and mercy, the spokesperson's message was that it is difficult for shame to even get a hold of someone who is seen for who they really are. A smaller but important difference: the description of the woman had changed from "the woman at the well who met Jesus," as the pastor wrote, to "the story about the woman at the well".

When asked about this practice in the first interview, the communications director in Järfälla smiled and said: "Oh, you caught on to me!" She continued to acknowledge that in a way she was doing the short summary as a translation to non-theologians in relation to her own experience of being interested in spirituality and theology, but "would probably not qualify as Christian, believing Christian, like pastors and people like that". Yet, she thought that Christian faith had been important to her, and that it could be important to others, if they are not excluded or turned away by a "churchy" language. In the final interview, she stated:

> What is my goal? It is to do a short summary, make accessible what [the sermon] is about, if possible in an unchurchy way [ . . . ] and I just try to formulate it without thinking too long or too deeply. In the beginning it just made me come into biblical formulations, and I thought that this will sound nice to the pastors, and to the people who do not read the Bible it is going to sound like gibberish. [laughs] And I thought about it: what am I trying to do? [ . . . ] If we are trying to meet as wide target audience as possible, then it is worth trying to not exclude the people who do not read the Bible.

Here we may note several interesting things. First, the spokesperson talks about her authority as role-based. According to her, she merely "packs" or "translates" the message. This is in line with the digital spokespersons described by Campbell, who emphasized

that their work was not about theological interpretation but about making the message of the church accessible. However, as shown in the example above, the reality is that the adaptation to a format and style that suits her understanding of contemporary secular culture and digital platforms is a theological interpretation in its own right. Furthermore, since this interpretation contextualizes the sermon to the listener, it contributes to the listener's own "auredit".

It is also important to note that her colleagues do not view her framing as mere translation but as collaboration and interpretation. When the rest of the team was asked about the digital spokesperson's work, they were very appreciative. For example, the other spokespersons (technicians) in Järfälla thought that "she writes amazing texts about the content of the worship service, that really encourages people to watch".

In sum: the spokespersons engaged in the preaching event through the framing of the delivered and edited sermon in social media. As shown above, they engage in many other ways including through editing, directing, altering of the sermon manuscript, enhancing, and contextualizing. The strategists' engagement is characterized by relational authority—authority as created and negotiated between different parties through communicative interaction in an interdependent relationship. The spokespersons' engagement is often characterized by relational authority. While their way of expressing themselves points to their being accustomed to wielding role-based authority, in practice they are often wielding a relational authority. How, then, might this engagement be understood?

## 5. How Can These Effects of RDC Engagement in the Preaching Event Be Understood?

### 5.1. It Can Be Understood as Co-Preaching

In relation to the homiletical discourse on polyphonic preaching, in which preaching is seen as a "dialogical, polyphonic co-authorship" that is created through various voices that supplement each other, I propose that the engagement of RDCs in the preaching event can be understood as co-preaching.

The engagement of the RDCs in the preaching event differs from the engagement in the roundtable conversations described by John McClure (1995), for example, where the preacher listens to others but ultimately serves as a curator of what should go into the sermon or not. In the conversational approach, the preacher is still very much in control.

In these two cases, however, the preacher/strategist no longer serves as a curator. The digital spokespersons have agency and authority in their own right, and it would be very difficult for the preacher/strategist to silence their voices if she wanted. This could explain why the strategists frequently mention the experience of loss of control. Notably, the loss of control is only mentioned as a problem in relation to the listeners who might misunderstand or scrutinize, not in relation to the spokespersons. On the contrary, they are regarded as gatekeepers who through their engagement described here will decrease the risk of misunderstandings (Mannerfelt and Roitto 2022b, pp. 74–75). The relational authority they practice and describe in their interviews also points towards something more than a regular conversational approach. This is mutual collaboration, a co-preachership.

Since the material for this case study was assembled within the framework of an action research project, the practitioners were introduced to the concept of RDCs and co-preaching and asked what they thought about this theory as an interpretation of their work. They all confirmed it, although one of the spokespersons, the communications director in Järfälla, was a bit reluctant at first. She again emphasized her role as translator who is only concerned with the form of the message, not the message itself. However, in the next interview two months later, she stated: "I think it was very interesting how we found out that what I'm writing—well, I feel that I have snuck into a preaching niche". The other RDCs confirmed that they indeed functioned as co-preachers; however the spokespersons' feelings about this were a bit ambiguous. They stated that it was empowering, but at the same time they recognized the stakes involved in this statement. As one of the spokespeople in the Järfälla team put it, half jokingly: the tack of co-preaching "is a great responsibility to put on three "morons". [like herself and her colleagues] Isn't that fatal?"

Though this digital form of co-preaching is new, co-preaching as practice is nothing new in the history of the Christian church. During late Antiquity and the Middle Ages, it was common for distinguished preachers to engage scribes who wrote down their sermons. The scribes also interpreted and edited the sermons, thus contributing to the content. Bernard of Clairvaux and Nicholas of Montiéramey are, together, an example of such "co-preaching" (de Gussem 2017, pp. 190–225). In some churches, it is an established practice to interrupt the preacher if the Holy Spirit encourages you to say something. The practice of call and response could also be called co-preaching (Crawford 1995; Richards-Greaves 2016; Thomas 2016). When it comes to the practice of framing the sermon, this also occurs in some communities, for example when the preacher is introduced or when the worship leader is praying for the preacher and/or the sermon before or afterwards. Sometimes the worship leader gives a summary of the sermon afterwards (Halldorf 2018, pp. 144–53).[13] Singing could also be seen as a practice of co-preaching, as is very evident in the revival movements.[14] As I noted above, the musician in Täby also made this parallel.

Furthermore, there might be others involved in a digitally mediated preaching event who could be considered co-preachers. As I have discussed elsewhere, people who comment on and share sermons in social media could be understood as co-preachers (Mannerfelt 2020, p. 209).

No matter who the co-preachers are, a consequence of co-preachership is that it may facilitate polyphony.

*5.2. It Can Be Understood as Enabling Polyphony*

The polyphonic preaching approach also aids in understanding how the co-preaching in these two cases, enabled as it is through digital media, effectively makes the preaching event more polyphonic. More voices contribute, including those of persons *and* mediational means. In these cases, it is not as with Rystad's preachers who started out with polyphony but fell into old patterns of "one single harmonized voice". The dialogical polyphony held up all the way, for better . . . or worse.

Indeed, it is worth asking what happens when the voices in the polyphony belong to people without formal theological education and who have not been ordained—or as the practitioners in Järfälla very pejoratively put it: "morons"—have such a large influence on the preaching event. Is there not a risk that the message of the gospel becomes contorted? As mentioned earlier, this is discussed by Sigmon (2017, pp. 178–80, 185–87), who in her homilecclesiological vision underlines the importance of preachers building up laypeople for the task of interpreting and communicating God's words and actions. The preachers' own preaching must model how the sacred texts and traditioned dogmas can be interpreted and communicated, while also helping them both to acknowledge their own authority and to discern when they are subordinated to unjust authority by others. Sigmon's proposition points to the importance of a concept like co-preaching, which reveals that spokespersons (and perhaps other types of RDCs) are actively partaking in the interpretation of the message of the church.

Finally, while the digitally mediated preaching events could be said to enable polyphonic preaching through the engagement of co-preachers, there are certain limitations to the polyphony, limitations that are caused by the very same digital mediation. In a discussion on liturgy in digital spaces, art historian Johannes Stückelberger points to the fact that the choice of camera angles and visual content affects how the words of the preacher are interpreted. This means that the pictures included in a digitally mediated worship service are not just contributing to an atmosphere, they are liturgical elements that create a dialogue with the sermon. However, at the same time as digital mediation enables new constellations of visualization and verbalization that would otherwise be impossible in the onsite church, it also delimits the listener's choice of which visual element will contribute to the interpretation. The listeners cannot let their gaze wander around the church and choose something else, for the camera and editor are directing it (Stückelberger 2021).

## 6. Summary

This case study of six digitally mediated preaching events in two Church of Sweden (CoS) Stockholm-area congregations aimed to describe and discuss what happens when the use of digital technology introduces new human actors into the preaching event. These actors were identified as "Religious digital creatives" (RDCs), a concept coined by Heidi Campbell in her study of religious authority and new media. The case study involved RDCs from the categories "digital strategist" (pastors and musicians) and "digital spokesperson" (technicians and communications directors).

The research questions that guided the study were "When and how do the RDCs engage in the preaching event?" and "How can these effects of RDC engagement in the preaching event be understood?" The source material consisted of observations, individual and focus group interviews, recordings of the services, and screen shots of how the recordings were presented when they were published on the congregations' websites and social media platforms. This material was analyzed with regard to their practices, how they understood those practices, and what kind of authority that emerged in those doings and sayings.

I found that the RDCs in this case study engaged in the preaching event in the verbalization phase, turning it into a phase of both verbalization *and* visualization. In addition, their engagement introduced a second phase of verbalization and visualization. More specifically, the RDCs engaged through editing direction, altering the sermon manuscript, enhancing, commenting, and framing. The RDCs' engagement was characterized by relational authority, that is authority created in negotiation through communicative interaction in a mutual and interdependent relationship.

The results was discussed in the light of the concept polyphonic preaching, that draws on the communication theory of Michail Bakhtin to describe preaching as dialogical and listeners as co-authors or even the primary authors of the sermon. In line with this, the RDCs were understood as co-preachers. The perspective of polyphonic preaching also shed light on how the practice of co-preaching increased and upheld the polyphony of voices that contributes to the dialogical character of the sermon.

**Funding:** This research received no external funding.

**Institutional Review Board Statement:** The study was approved by the Swedish ethical review authority and conducted in accordance with their guidelines.

**Informed Consent Statement:** Informed consent was obtained from all subjects involved in the study.

**Data Availability Statement:** The data are not publicly available due to privacy.

**Conflicts of Interest:** The author declares no conflict of interest.

## Notes

[1] "Preaching event" is a concept that has become increasingly common in homiletical discourse. But just like the term "practice of preaching" (Rystad 2020, p. 19), it is not always clear what is meant by "preaching event," since the term has been used in a variety of ways over time. John Claypool (1980), one of the first to invoke the term, understands a preaching event as the event when the human utterances of the preacher become God's living words to the listeners. To other homileticians, it designates the situation (oral event) when the preacher is speaking to the listeners (Bruce 2013; Maddock 2017). The concept preaching event could also be understood as the event that occurs in performative situations where the preacher, listeners and message interact (Fahlgren 2006, pp. 43–47). Finally, the concept could also be employed like Wilfrid Engemann, who argues that since it is not entirely clear when the sermon actually becomes a sermon, it is important to keep all the parts of the process together in the analysis. The concept "preaching event" is therefore used to designate everything from the preparation phase to the moment the audience listens (Engemann 2019, pp. xix–xx). In this article, I draw on Engemann's broad understanding of the concept. As Linn Sæbø Rystad has argued, there are several benefits to conceiving of preaching as an event. It allows for understanding preaching as a practice, which in turn sheds light on how preaching is both "processual, performative and emerging" and radically relational. The term also highlights the importance of material entities like architecture, art, artifacts, and other visual aids for the meaning-making process (Rystad 2020, pp. 122–23).

[2] In this article, I use the concept actor in the same sense as it is used Bruno Latour (1999, pp. 120–23) in his Actor-Network Theory: an actor is something that acts or to which activity is granted by others.

3    *Church in Digital Space* was a collaboration between the CoS diocese of Stockholm and University College Stockholm. It involved five researchers and seven congregations, and it was led by professor of practical theology Jonas Ideström and the bishop's theological advisor, Sara Garpe. The project took place between January 2021–September 2022 (Garpe et al. 2022). The research project was inspired by Theological action research (TAR) (see for example Watkins 2020), as well as the methods and concepts developed by Jonas Ideström (2015) and the bishop of the CoS diocese of Stockholm, Andreas Holmberg (2019). There are very few action research projects that focuses on homiletics, Boyd (2018) being an exception.

4    As stated in the introduction to the research report, the starting point of action research consists of two ideas: (1) Research can contribute to solving actual problems and develop knowledge and skills; (2) Participants possess knowledge that could significantly contribute to the research process. Research is therefore carried out in collaboration between researchers and practitioners, who come together in a process of interpretation and reflection. Practitioners contribute with their experience and knowledge, while researchers contribute resources like methods, theories, and research from other contexts (Garpe et al. 2022, pp. 13–16).

5    For a detailed description of how the source material was created, see (Mannerfelt and Roitto 2022a).

6    Although not its primary purpose, the article does contribute to confirming Campbell's hypothesis by giving concrete examples of how religious institutions are, in fact, changed by their adoption of new media.

7    In the following, I will refer to them with professional title and congregation, for example "pastor in Täby" or "technician in Järfälla".

8    Before Lorensen, there are other Scandinavian theologians who have used Bakhtin in their homiletical reflection. See for Karlsson (2008), and Bjerg and Lynglund (2010). In addition, it is not just Scandinavian homileticians who have drawn on Bakhtin. See Harris (2004).

9    Homileticians suggest various reasons for this. It could be a matter of hermeneutics—for example historical-critical Bible studies who challenged the idea of the Biblical text as a source of absolute truth and authority—or general societal developments like secularization, pluralism, or postmodernism (Mervin 1983; Brueggemann 1990; Tornfelt 2004; Davies 2007). Interestingly, digitalization is never mentioned as a reason for changes in the understanding of authority.

10   Here we may notice that Engemann's model is an example of the phenomenon I discussed in the introduction. While he does acknowledge that materiality plays a role in the interpretation as part of the particular context, it tends to tread into the background as part of the overall context. The main foci in his homiletical discussion are the Bible text, the preacher and the listener, and consequently the phase is conceived as centered on words.

11   This is briefly discussed also in the Evangelical Lutheran Danish Church (ELDC) research report on worship services during the COVID-19 pandemic. The choice of location in the local church space has theological significance. It also enables perspectives that would not normally be possible for the participant, like being face-to-face with a preacher who stands in the pulpit, or a bird's-eye view from the church ceiling. (*Når Folkekirken Skal Spille efter Reglerne—Men Uden for Banen Folkekirkens Håndtering af Coronaperioden i Foråret 2020* 2020, pp. 177–85).

12   "B-roll" is a term used in film production that designates supplemental or alternative footage that intercut with the main shot.

13   It is interesting to see that these kinds of introductions and summaries to elements of the services are practices that congregations tend to cut out in digital worship services (See Mannerfelt 2021, p. 100).

14   One of the first and most well-known examples of a preacher-singer collaboration where the singer act as a sort of co-preacher is Dwight L. Moody and Ira D. Sankey. In the Swedish context there are several examples, like Nelly Hall and Ida Nihlén (Gunner 2003) and Lewi Pethrus and Einar Ekberg (Halldorf 2017, p. 216).

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
