# Peer review of "Co-Preaching: The Effects of Religious Digital Creatives’ Engagement in the Preaching Event"

_religions, doi:10.3390/rel13121135_

Round 1

Reviewer 1 Report

This is a much-needed action research study in the field of digital ecclesiology. This essay lays the foundation for future research. Aside from a handful of typos (i.e., lines 135, 144) and a question about the singular mention of Levinas (lines 325-326), this is an excellent essay.

Author Response

The reviewer suggests a thourough proof reading, but otherwise she seems content with the article as it is. 

In the revision of the article, the typos have been corrected, as well as the singular mention of Emmanuel Levinas. 

I thank the reviewer for her time and effort. 

Reviewer 2 Report

All of my comments and suggested edits are in the PDF attached.  This is a compelling paper that is timely, relevant to the field of homiletics, and makes a significant contribution to homiletic theory in the digital age.  In my comments I pointed out some things that could be addressed that would strengthen the argument and clear up some possible points of confusion.

Author Response

I see that the reviewer has made a very close reading, and noted typos and where the text could benefit from clarifications. I really appreciate that the reviewer has taken time to do such a thourough reading! I have revised the article accordingly. 

The reviewer has made her comments in the attached file, and here I will comment on the points that raise issues other than just clarifications and typos.  

Point 1 The reviewer thinks that it is confusing to introduce the concept of “co-preaching” in section 2.1, since the concept has not yet been defined. It is a very good point, and I have therefore removed that sentence.

Point 2 In relation to section 2.1, the reviewer asks if Campbell notes that an individual can be in more than one category. She points out that in smaller congregations the pastor has to occupy two or three roles. As the reviewer notices in her next comment, this question is addressed already in the next section of the text. I have considered R2’s suggestion to mention it earlier for clarity, but since the other two reviewers have not mentioned it as a particular problem, I think it can stand as it is.

Point 3 The reviewer suggests in her comment that the term co-preacher should not be used until it has been properly explained. She writes: “At this point in the article, I am not convinced this is an accurate or appropriate term to use in describing RNCs.  This may be because the term "co-preacher" has not yet been defined, so I don't know how you're thinking about this role.” This is a good point, and I have therefore removed it.

In the same comment she continues: “This means that even the word "preacher" needs to be defined. How do you understand the act of preaching? Whose definition are you using? I think this is important to establish.” The answer to this question is found the very first sentences of the article where I write: “As Wilfrid Engemann (2019, pp.3-4) has shown, the preaching event is a process of comprehension and communication that consists of several phases of text interpretations and text introductions that involve the interaction between the authors of the Bible text, the Bible, preacher, sermon manuscript, the delivered sermon, listener, and the “auredit” (what the listener has heard), each in their specific context.” In the beginning of the discussion of results in section 3, I return to Engemann’s definition and explain it with a figure. I considered referring to Engemann’s definition a third time here, but since it seems to be the use of the concept “co-preacher” that actualizes the need for a reminder of the definition here, I think the problem for the reader is solved as I have deleted the sentence.

Point 4 The reviewer comments that Gaarden and Lorensen’s concept “Listeners as authors” does not make sense to her, and she would like to see some critical engagement with the idea. I have rewritten the section in order to show how the authors reach this conclusion. Hopefully this will clarify. 

Point 5 The reviewer suggests that I refer to McClure’s book The Roundtable Pulpit. An excellent suggestion, and the reference has been added.

Point 6 The reviewer comments the section where I paraphrase Sigmon’s vision of homilecclesiology and asks:

"Hm, then what is the point of preaching at all? How can preaching avoid truths and timeless statements?  Or is the problem the notion of "imposition"? I have not had a chance to read Sigmon's book, so I'm not sure if this paraphrase accurately reflects her thinking.  But if it does, I think you'll need to critically engage this in light of the preacher's role as the theologically-trained person in this relationship." 

As I read Sigmon, her point is (as R2 suspects) not to say that there are no timeless truths (indeed, she emphasize their importance!), but the problem is when they are imposed on the listeners. She has also written about the relationship between the preacher (as being the theologically trained person) and the listeners. I refer to this at the end of the article, but it is beneficial to include it here as well. I have rewritten the section to clarify her argument.

Point 7 In section 3, the reviewer lifts an important question: “Isn't there verbalization and visualization in the traditional, onsite preaching event as well?  Where is the performative and visual aspect of preaching accounted for in Figure 1?” She also asks what the difference might be between the verbalization/visualisation going on in onsite preaching, and the verbalization/visualisation in online preaching. This was a particularly helpful comment, and I am very grateful that R2 brought this up.

I have rewritten the section to point out that while this phase always was about both verbalization and visualization, you may argue that the visual is emphasized in digital culture. I have also added a endnote about how Engemann's model is an example of the phenomenon I discussed at the beginning of the article, that homileticians tend to focus on the interaction between preacher, listener and the Bible (words) and tends to forget the importance of materiality. I am thankful for this insightful question of the reviewer! 

Point 8 In the beginning of section 4, the reviewer comments: “In this section, I'm not seeing what role digital entrepreneurs play. They are listed above as one of the 3 categories of RDCs, but where are they in these case studies?” This question points to the fact that the mentioning I do in the methodology and material section about this (that the RDCs considered in this study are only spokespersons and strategists, not entrepreneurs) may not be sufficient. I have therefore added a reminder of this in the beginning of section 4.

Point 9 At the end of section 4.2, the reviewer comments that “All of this is very positive.  I'm wondering if there are any downsides to this co-authoring, such as an end result that either obscures or distracts from the sermon, for example.  Also, I'm wondering how authority is negotiated when there are disagreements between the spokesperson and strategist.  Like, does the preacher have final say? Or does the spokesperson/technician because they do the final edit?”

In these two cases, there were no instances when the spokespersons’ b-roll imagery obscured the strategists’ words. The collaboration was very thoughtful and respectful. However, as R2s question points out, there might be a risk that this happens in other cases. I have added a comment about this.

Disagreements were solved through negotiation, as described in the example with the All Saints sermon. I have added a sentence that points this out.

Point 9 The reviewer comments in the section 5.2 “I'm not seeing Sigmon's response really speaking to the problem of a contorted interpretation of the gospel by co-preachers. You're putting forth a viewpoint that is perhaps aspirational, because the fact is that interpretations can not only be contorted, but also harmful.  So what is the preacher's role in ensuring against possible theological malpractice in this process?”

As mentioned in my response to point 6, I have clarified Sigmon's argument on this issue.

I have attached the file if the reviewer would like to have a look at the revisions and see if she judges them sufficient.

Again, I would like to express gratitude for the reviewers time and effort. 

Reviewer 3 Report

I appreciate this text as very erudite, of high quality in terms of content and form. I have almost no critical comments about it - except for one methodological one.

First of all, it should be emphasized that the author works excellently with literature. His references to previous works are not only formal and enumerative, but he works with them intensively and meaningfully. He relies primarily on two key authors, giving their work a new dimension: H. Campbell and M. Bakhtin. He presents an interesting application of Bakhtin's polyphony, speaking of "polyphonic author-position" (l. 255). He refers to the Scandinavian homiletic tradition linked to Bakhtin's theory and considers the "pulpit-pew binary", the authority of the preacher and its degree of relativity.

H. Cambell is the author of key theses on religion in the digital environment and on "digital religion". The author takes from her a very apt division of "religions digital creatives" (RDCs) into three groups: entrepreneurs, spokespersons and strategists. Subsequently, he applies the second and third categories in his own research. However, he also points to an interesting contradiction between Campbell's findings before the covid pandemic and his findings: Campbell noticed something like a contradiction: the same institutions that hired the RDCs to do digital media work are reluctant about the use of digital technology. But this author claims that after Covid-19 this is not the case and "spokespersons" and "strategists" cooperate very well and willingly.

The author argues for an important thesis in his study: that RDCs have the function of "co-preachers". He examines, "what kind of authority they perform through their words and actions" (l. 155). He claims that by editing the sermon, they enter into its perception, because the audience no longer interprets the spoken sermon, but the edited sermon. However, their work is not about theological interpretation but about making the message of the church accessible (l. 673). Finally, the author asks the important question "what happens when the voices in the polyphony belong to people without formal theological education and who have not been ordained have such a large influence on the preaching event" (l. 758 - 760).

Considering the above, I find this study very up-to-date, well-crafted, and useful for scientific research in the given area. My critical comment is about methodology and language proofreading. The author states that he worked with individual and focus group interviews - but he does not describe his methodology in more detail. It is necessary to dedicate a separate part of the article to the presentation of his specific procedures: what method he used, why it was the most suitable, what people he interviewed, how many individual interviews there were, how many focus groups, how they were organized, how the recordings were worked with... A thorough language proofreading is also necessary because the text contains several typographical errors.

Author Response

I was happy to see that the reviewer seems overall content with the article and only has two requirements.

First, a language proofreading. This problem has been addressed in the revised version of the article.

Second, a description of methodology. The revised version of the article now contains a section that describes the processes of gathering and analyzing material, and the character and quantity of the material (section 2). I will attach the revised article here, in case the reviewer would like to see if the revision is sufficient and meets her requirements.

Finally, I would like to thank the reviewer for her time and effort! 
